# Characterization and genomic analysis of the Lyme disease spirochete bacteriophage φBB-1

**Dominick R. Faith[1]☉, Margie Kinnersley[1]☉, Diane M. Brooks[1], Dan Drecktrah[1], Laura S. Hall[1], Eric Luo[2], Andrew Santiago-Frangos[3], Jenny Wachter[2], D. Scott Samuels[1], Patrick R. Secor[1] \***

**1** Division of Biological Sciences, University of Montana, Missoula, Montana, United States of America,
**2** Vaccine and Infectious Disease Organization, Saskatoon, Canada, **3** Department of Biology, University of Pennsylvania, Philadelphia, Pennsylvania, United States of America

☉ These authors contributed equally to this work.
\* Patrick.secor@mso.umt.edu

**Data Availability Statement:** All data are available in the manuscript and supplemental files. Sequencing data have been submitted to NCBI, PRJNA1059007.

## Abstract

Lyme disease is a tick-borne infection caused by the spirochete *Borrelia* (*Borreliella*) *burgdorferi*. *Borrelia* species have highly fragmented genomes composed of a linear chromosome and a constellation of linear and circular plasmids some of which are required throughout the enzootic cycle. Included in this plasmid repertoire by almost all Lyme disease spirochetes are the 32-kb circular plasmid cp32 prophages that are capable of lytic replication to produce infectious virions called φBB-1. While the *B. burgdorferi* genome contains evidence of horizontal transfer, the mechanisms of gene transfer between strains remain unclear. While we know that φBB-1 transduces cp32 and shuttle vector DNA during *in vitro* cultivation, the extent of φBB-1 DNA transfer is not clear. Herein, we use proteomics and long-read sequencing to further characterize φBB-1 virions. Our studies identified the cp32 *pac* region and revealed that φBB-1 packages linear cp32s via a headful mechanism with preferential packaging of plasmids containing the cp32 *pac* region. Additionally, we find φBB-1 packages fragments of the linear chromosome and full-length plasmids including lp54, cp26, and others. Furthermore, sequencing of φBB-1 packaged DNA allowed us to resolve the covalently closed hairpin telomeres for the linear *B. burgdorferi* chromosome and most linear plasmids in strain CA-11.2A. Collectively, our results shed light on the biology of the ubiquitous φBB-1 phage and further implicates φBB-1 in the generalized transduction of diverse genes and the maintenance of genetic diversity in Lyme disease spirochetes.

## Author summary

Lyme disease is a tick-borne disease caused by the bacterium *Borrelia (Borreliella) burgdorferi*. *Borrelia* bacteria have complex genomes that include various circular and linear DNA plasmids. Horizontal gene transfer occurs between Lyme disease bacteria; however, the mechanisms are unclear. A key component of the *Borrelia* genome is the 32-kb circular plasmid prophage cp32. When cp32 prophages are induced, infectious virions called

**Funding:** PRS is supported by NIH grants R21AI151597 and P30GM140963. MK is supported by NIH grant P20GM103474. DRF is supported by NSF GRFP grant 366502. A.S-F. is a M. Jane Williams and Valerie Vargo Presidential Assistant Professor of Biology and is supported by NIH grants K99GM147842 and R00GM147842, and by the Postdoctoral Enrichment Program Award from the Burroughs Wellcome Fund (G-1021106.01). The funders had no role in study design, data collection and analysis, decision to publish, or preparation of the manuscript. The authors declare no conflicts of interest.

**Competing interests:** The authors have declared that no competing interests exist.

φBB-1 are produced. It is thought that φBB-1 phages horizontally transfer DNA between Lyme disease bacteria. Using proteomics and long read DNA sequencing, we found that φBB-1 virions package not only cp32 plasmids but also fragments of the bacterial chromosome and other plasmids. Additionally, our sequencing revealed unique features of the packaged DNA, such as the pac site that is used to initiate DNA packaging into φBB-1 capsids. These findings implicate a role for φBB-1 in horizontal gene transfer between *Borrelia* strains, contributing to their genetic diversity. Understanding this process is vital for developing better strategies to combat Lyme disease.

## Introduction

The bacterium *Borrelia* (*Borreliella*) *burgdorferi* is the causative agent of Lyme disease, the most common tick-borne disease in the Northern Hemisphere [1–3]. Lyme disease spirochetes have complex and highly fragmented genomes composed of a ~900-kb linear chromosome and up to twenty distinct and co-existing linear and circular plasmids that are similar but not identical across the genospecies [4–6].

As a vector-borne pathogen, *B. burgdorferi* relies on the differential expression of several outer surface lipoproteins to transmit from its tick vector to a vertebrate host [7]. As such, a large fraction of the *B. burgdorferi* genome encodes outer membrane lipoproteins, mostly carried on the plasmids [6,8,9].

In natural populations, genetic variation in outer membrane lipoprotein alleles is associated with species-level adaptations [6,8–10] and variation in outer membrane lipoprotein alleles across the genospecies is driven primarily by horizontal gene transfer [5,11–21]. However, the mechanism(s) by which heterologous *B. burgdorferi* strains exchange genetic material are not well defined.

Viruses that infect bacteria (phages) are key drivers of horizontal gene transfer between bacteria [22]. The genomes of nearly all sequenced Lyme disease spirochetes include the 32-kb circular plasmid (cp32) prophages (**Fig 1A and 1B**) [4]. The cp32s carry several outer membrane lipoprotein gene families including *mlp* and *ospE/ospF/elp* (*erps*), which are all involved in immune evasion [23–27] and exhibit sequence variation that is consistent with historical recombination amongst cp32 plasmid isoforms [20,21,28]. Recent work indicates that cp32

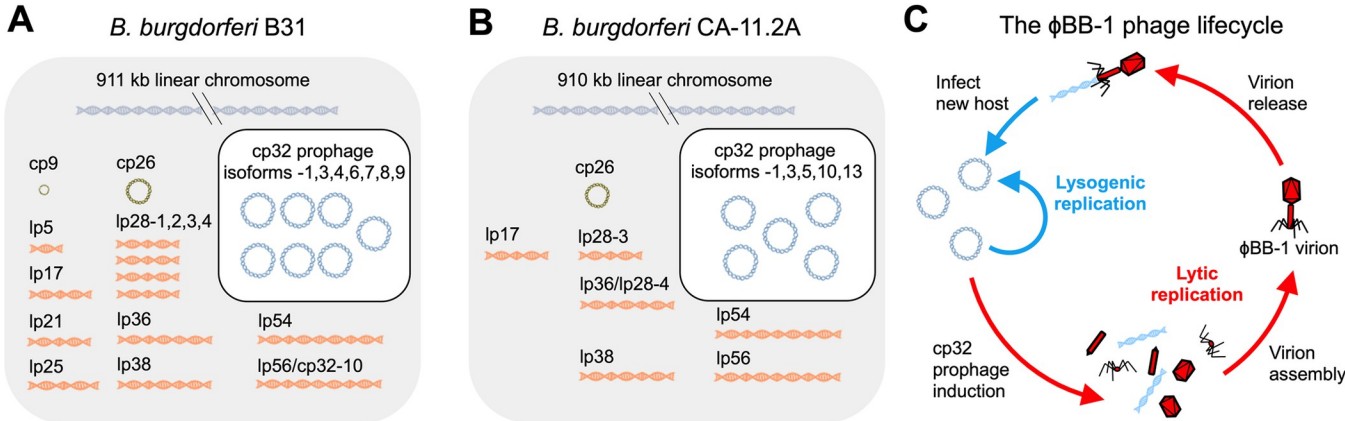

**Fig 1. The *B. burgdorferi* genome is highly fragmented and is composed of a linear chromosome, linear and circular plasmids, and cp32 prophages.** The genomes of *B. burgdorferi* strains (**A**) B31 and (**B**) CA-11.2A are shown. (**C**) The temperate φBB-1 phage lifecycle is depicted.

prophages are induced in the tick midgut during a bloodmeal [9,29,30]. When induced, cp32 prophages undergo lytic replication where they are packaged into infectious virions designated ϕBB-1 (**Fig 1C**) [31–33].

In addition to horizontally transferring phage genomes between bacterial hosts (transduction), phages frequently package and horizontally transfer pieces of the bacterial chromosome or other non-phage DNA (generalized transduction) [34]. Generalized transduction was first observed in the *Salmonella* phage P22 in the 1950s [35] and since then has been observed in numerous other phage species [34,36–39]. ϕBB-1 is a generalized transducing phage that can horizontally transfer shuttle vectors carrying antibiotic resistance cassettes between *B. burgdorferi* strains [31,40]. However, to our knowledge, generalized transduction of anything other than engineered plasmids by ϕBB-1 has not been observed.

Here, we define the genetic material packaged by ϕBB-1 virions isolated from *B. burgdorferi* strain CA-11.2A. Our proteomics studies confirm that ϕBB-1 virions are composed primarily of capsid and other phage structural proteins encoded by the cp32s; however, putative phage structural proteins encoded by lp54 were also detected. Long-read sequencing reveals that ϕBB-1 virions package a variety of genetic material including cp32 isoforms that are linearized at a region immediately upstream of the *erp* locus (*ospE/ospF/elp*) and packaged into ϕBB-1 capsids via a headful genome packaging mechanism at a packaging site (*pac*). When introduced to a shuttle vector, the *pac* region promotes the packaging of shuttle vectors into ϕBB-1 virions, demonstrating the utility of ϕBB-1 as a tool to genetically manipulate Lyme disease spirochetes. Additionally, full-length contigs of cp26, lp17, lp38, lp54, and lp56 are recovered from packaged reads as are fragments of the linear chromosome. Finally, long-read sequencing of packaged DNA allowed us to fully resolve most of the covalently closed hairpin telomeres in the *B. burgdorferi* CA-11.2A genome.

Overall, this study implicates ϕBB-1 in mobilizing large portions of the *B. burgdorferi* genome, which may explain certain aspects of genome stability and diversity observed in Lyme disease spirochetes.

## Results

### ϕBB-1 phage purification, virion morphology, and proteomic analysis

In the laboratory, lytic ϕBB-1 replication (**Fig 1C**) can be induced by fermentation products such as ethanol [40,41]. We first measured ϕBB-1 titers in early stationary-phase cultures (~$1 \times 10^8$ cells/mL) of *B. burgdorferi* B31 or CA-11.2A induced with 5% ethanol, as described by Eggers *et al.* [40]. Seventy-two hours after induction, bacteria were removed by centrifugation and filtering. Virions were then purified from supernatants by chloroform extraction and precipitation with ammonium sulfate. Purified virions were treated for one hour with DNase to destroy DNA not protected within a capsid and treated with chloroform to inactivate DNase; quantitative PCR (qPCR) was then used to measure packaged cp32 copy numbers.

*B. burgdorferi* strain CA-11.2A consistently produced ~10 times more phage than B31 (**Fig 2A**) and was selected for further study. Imaging of purified virions collected from CA-11.2A by transmission electron microscopy reveals virions with an elongated capsid and contractile tail (**Fig 2B**), which is similar to the Myoviridae morphology of ϕBB-1 virions produced by strain B31 *in vitro* [9,42,43] and by a human *B. burgdorferi* isolate following ciprofloxacin treatment [44].

Mass spectrometry analysis of purified virions identified ten capsid and other structural proteins encoded by the cp32s including the major capsid protein and capsid fibers (**Fig 2C and S1 Table**). We also detected highly conserved predicted phage capsid proteins encoded by lp54 (**Fig 2D**). Of note, the highest abundance proteins detected were OspC, OspA, and

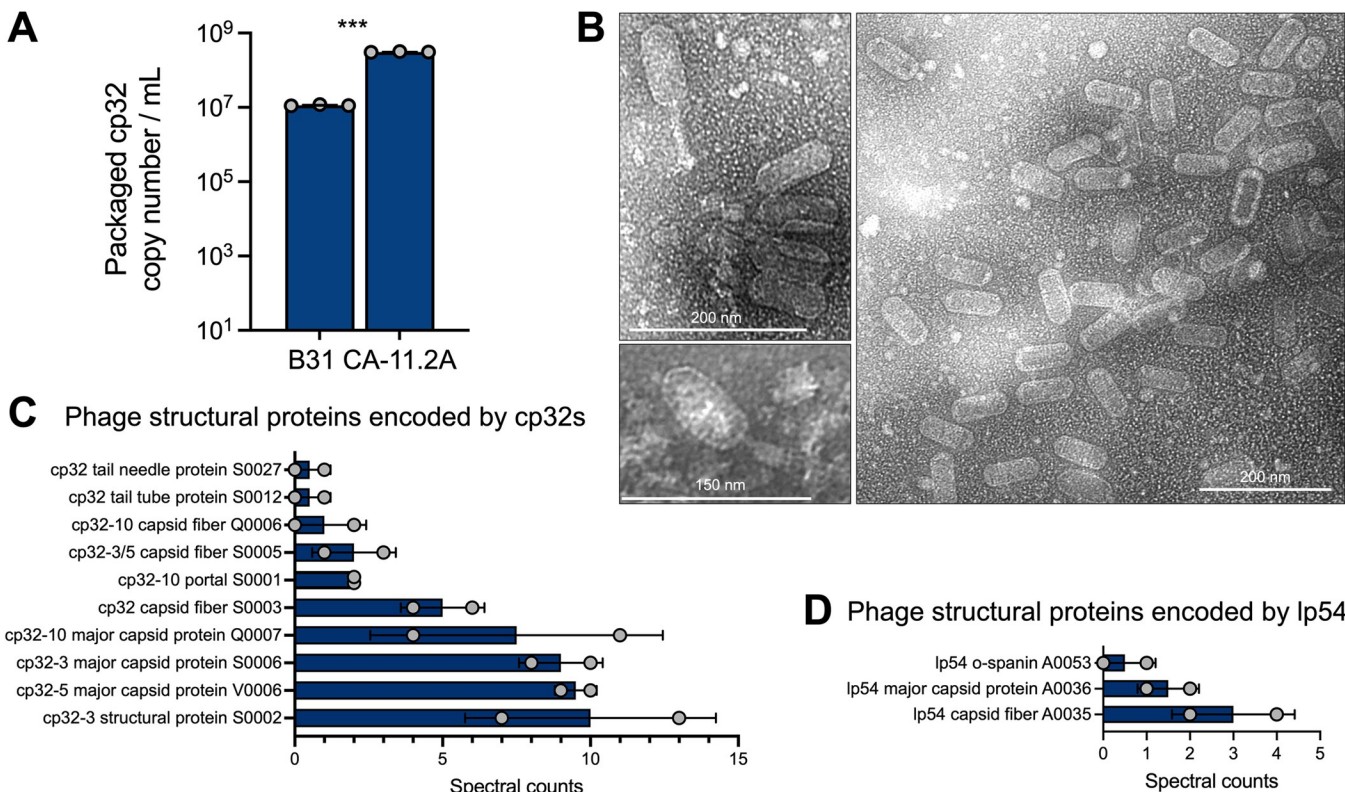

**Fig 2. φBB-1 phage titer, virion morphology, and proteomic analysis. (A)** Packaged, DNase-protected cp32 copy numbers in bacterial supernatants were measured by qPCR. Data are the SE of the mean of three experiments, ***$p < 0.001$. **(B)** Virions were purified from 4-L cultures of *B. burgdorferi* CA-11.2A and imaged by transmission electron microscopy. Representative images from two independent preparations are shown. **(C and D)** HPLC-MS/MS-based proteomics was used to identify proteins in two purified virion preparations. The SE of the mean of spectral counts for peptides associated with the indicated phage structural proteins are shown for each replicate. See also **S1 Table** for the complete proteomics dataset.

GroEL, which dominate the *B. burgdorferi* proteome and are known contaminants in protein samples [45,46]. While the virions we visualized all appear to have the same elongated capsid morphology, virions with a notably smaller capsid morphology have been isolated and imaged from *B. burgdorferi* CA-11.2A [31]. These observations raise the possibility that there are multiple intact phages inhabiting the CA-11.2A genome.

## φBB-1 virions package portions of the *B. burgdorferi* genome

We performed long-read sequencing on DNA packaged in purified φBB-1 virions, as outlined in **Fig 3**. Although intact *B. burgdorferi* cells were removed via both centrifugation and filtration prior to chloroform treatment, there is concern that contaminating unpackaged *B. burgdorferi* chromosomal or plasmid DNA co-purifies with phage virions. To control for this, we spiked purified φBB-1 virions with high molecular weight (>20 kb) salmon sperm DNA (**Fig 4A**) at 1.7 μg/mL, a concentration that approximates the amount of DNA released by $3 \times 10^8$ lysed bacterial cells into one milliliter of media [47]. Samples were then treated with DNase overnight followed by phage DNA extraction using a proteinase K/SDS/phenol-chloroform DNA extraction protocol [32]. Purified DNA was directly sequenced using the Nanopore MinION (long read) platform.

Across three replicates, we recovered a total of 110,986 nanopore reads >700 bp in length that met a minimum q-score threshold of 7. Kraken [48] and BLAST analyses indicated that

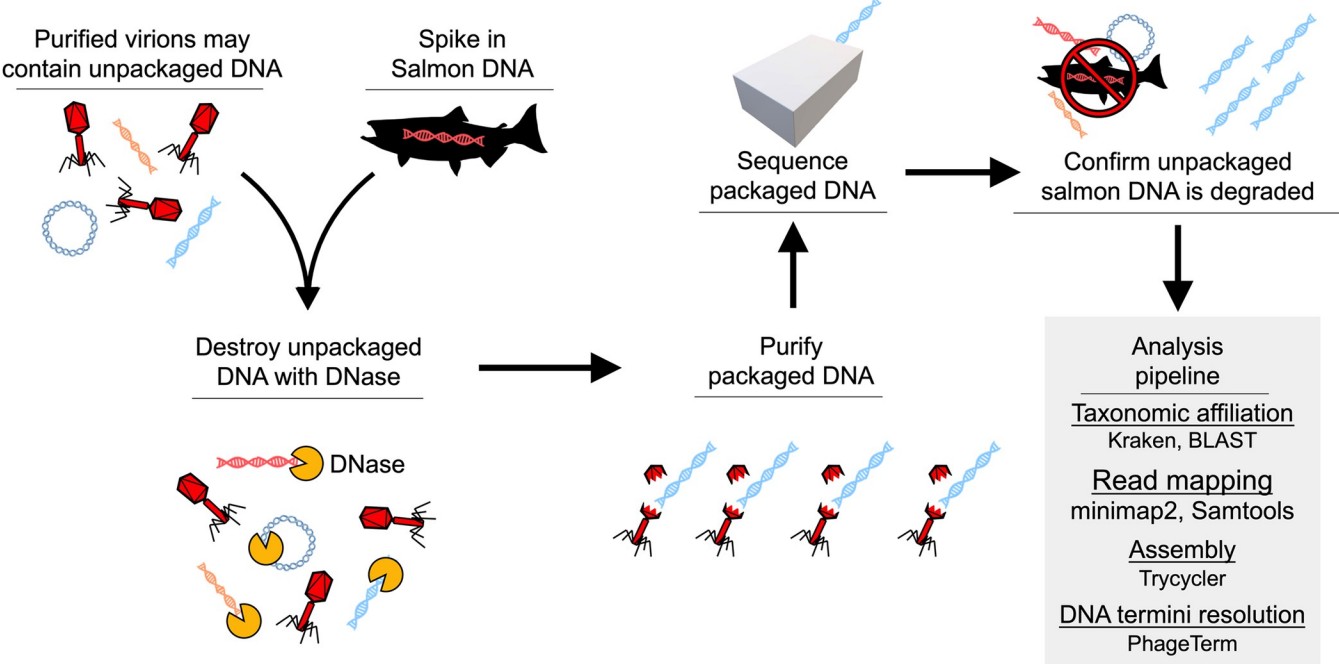

**Fig 3. Workflow for sequencing packaged φBB-1 DNA.**

the DNase treatment successfully degraded unpackaged DNA, as only 155 reads (0.14% of the total) with an average length of 1.2kb were derived from the salmon-sperm DNA spike-in (**Fig 4B**). To further reduce the possibility of unpackaged *B. burgdorferi* DNA carryover, we imposed a stringent 5kb read-length cutoff, thus reducing the number of salmon-derived reads to zero and leaving a total of 58,399 reads (**Fig 4C**) with a median length of ~12.3 kb

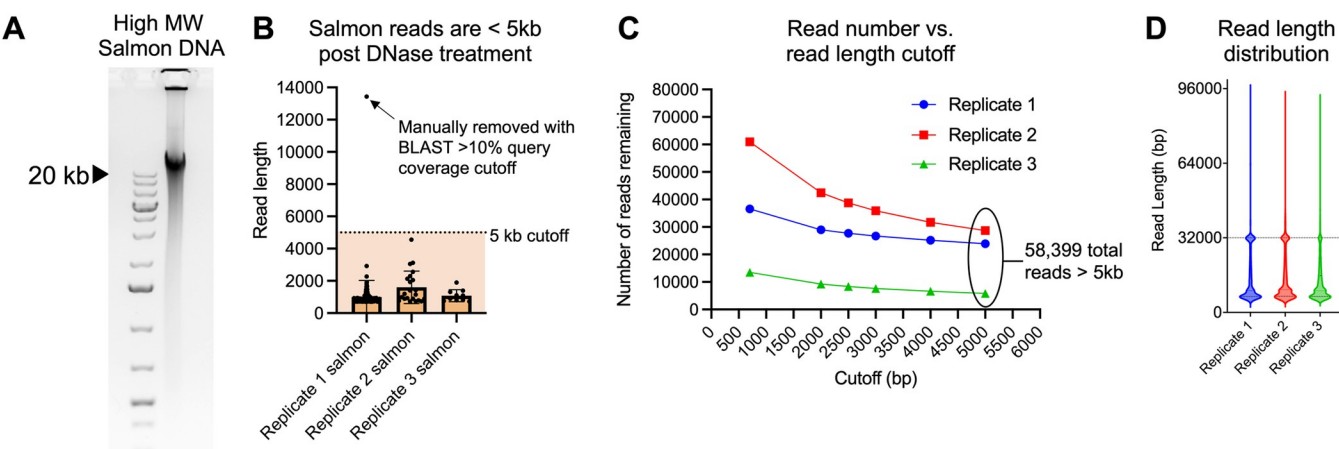

**Fig 4. Establishing a 5kb read length cutoff to exclude unpackaged reads.** (**A**) The salmon sperm DNA used to spike purified phages prior to DNase treatment was run on an agarose gel to estimate its size. Note that the majority of salmon DNA is larger than the 20-kb high molecular weight marker in the left lane. (**B**) 0.14% of 110,986 reads > 700 bp, 0.14% were classified as matching salmon sequences. Reads classified as salmon were plotted as a function of their length for each replicate. Error bars represent the SE of the mean of three replicate experiments. All reads except one (arrow) were below 5 kb in length (dashed line) with an average length of 1.2 kb. (**C**) Read length cutoff was plotted as a function of the number of reads remaining in each replicate dataset. In total, 58,399 reads remain after establishing a 5-kb cutoff. (**D**) Read length for all reads >5 kb in each replicate was plotted.

(Fig 4D). Note that we detected a high number of ~32 kb reads in each replicate which are the approximate size of cp32 prophages (Fig 4D, dashed line).

Overall, ~99.6% of packaged reads >5 kb were classified as *B. burgdorferi* (Fig 5A), the majority of which (~79%) were cp32 isoforms (Fig 5B). Cp32-10 and cp32-3 were preferentially packaged (~32% and ~25%, respectively) followed by cp32-13 and cp32-5 (each at ~10%) (Fig 5B). Reads mapping to cp32-3, cp32-5, cp32-10, and cp32-13 had a mean coverage of over 1,000× (Fig 5C). Cp32-1 reads accounted for only about one percent of all packaged reads (Fig 5B) and had lower mean coverage of approximately 36× (Fig 5C), suggesting that cp32-1 was not undergoing lytic replication. Read length distributions across cp32s indicate that full-length ~32 kb molecules were often recovered for cp32-3, cp32-5, and cp32-13, but less frequently for cp32-1 and cp32-10 (Fig 5D). The maximum read lengths recovered for each element in the CA-11.2A genome are listed in S2 Table.

Additionally, 11.6% of reads > 5 kb mapped to the linear chromosome and ~6.3% of reads >5 kb mapped to lp54 (Fig 5B). The remaining reads mapped to all the defined genetic elements of *B. burgdorferi* CA-11.2A including plasmids cp26, lp17, lp36/lp28-4, lp38, lp56, and lp28-3 at 1–2% each (Fig 5B). The average length for packaged chromosome reads and other low-frequency plasmids packaged by φBB-1 is ~8,000 bp (Fig 5D). This may indicate a packaging bias towards shorter DNA molecules for non-cp32 DNA molecules. *De novo* assembly of packaged reads produced full-length contigs of all cp32s, lp17, cp26, lp36, lp38, lp54, and lp56 (S1 Fig), suggesting that full-length versions of these plasmids are packaged by φBB-1.

Of note, the CA-11.2A genome was reported to contain a unique plasmid, lp36/lp28-4, that is thought to have arisen from the fusion of lp36 with lp28-4 [49]. *De novo* assembly of packaged reads resolved lp36/lp28-4 into individual lp36 and lp28-4 contigs (S1E and S1F Fig).

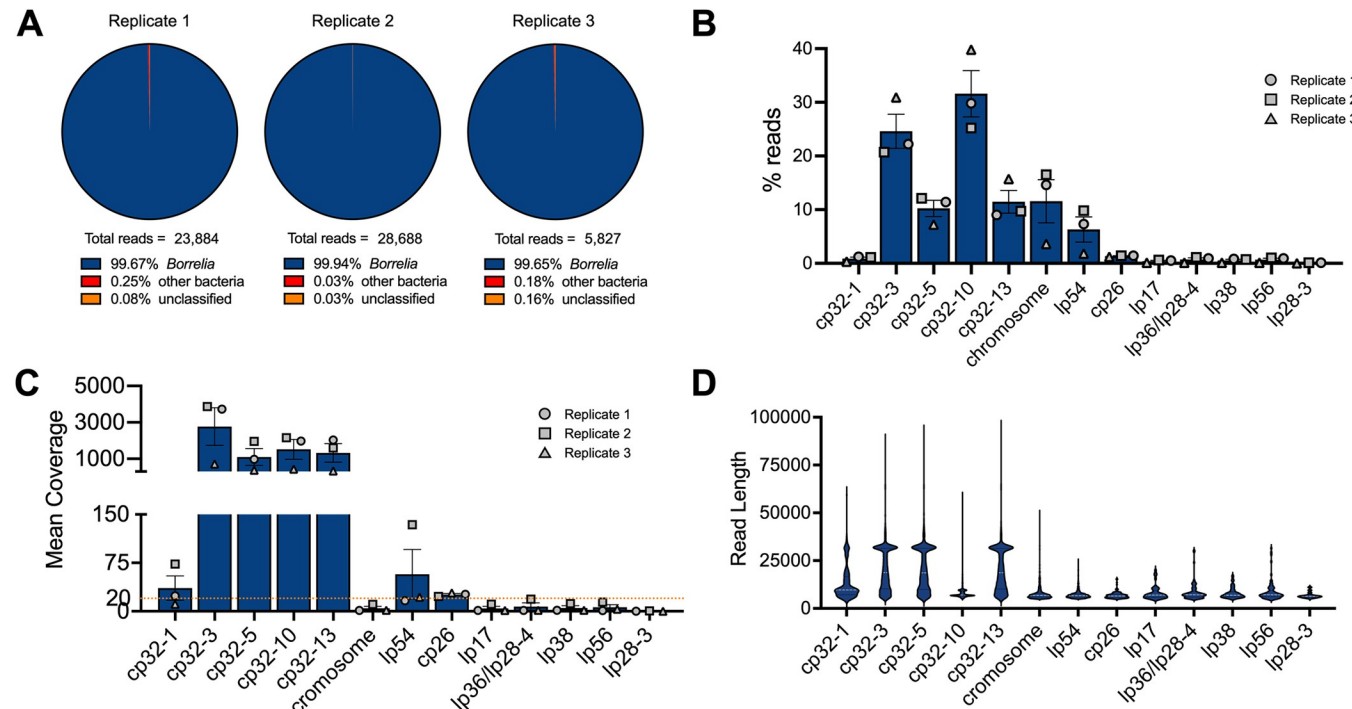

**Fig 5. φBB-1 virions package cp32 isoforms, chromosome fragments, lp54, and other plasmid DNA. (A)** Kraken and BLAST were used to determine the taxonomic affiliation of reads >5kb. Note that no eukaryotic reads were identified. **(B and C)** The (B) percent and (C) mean coverage for reads affiliated with the indicated *B. burgdorferi* plasmid or linear chromosome are shown for each replicate. Error bars represent the SE of the mean. **(D)** Read length distributions for the indicated plasmids or chromosome are shown.

Additionally, whole genome sequencing (MiSeq) of our CA-11.2A strain confirmed that lp36 and lp28-4 are separate as no reads that span the lp36-lp28-4 junction were observed and coverage depth was notably different between lp36 and lp28-4 (~200× vs. 25×, respectively, **S2A Fig**). Furthermore, PCR confirmed the sequencing results (**S2B–S2D Fig**). These data indicate that the lp36/lp28-4 plasmid is two distinct episomes in our CA-11.2A strain.

Collectively, these results indicate that in addition to cp32 molecules, φBB-1 is capable of packaging non-cp32 portions of the *B. burgdorferi* genome. We discuss the major packaged DNA species in the following sections.

## cp32 molecules are linearized near the *erp* locus and packaged via a headful mechanism

Our sequencing data provide insight into how φBB-1 packages cp32 molecules. Many phage species package linear double-stranded DNA genomes that circularize after being injected into a host [50]. Because DNA isolated from φBB-1 virions is thought to be linearized [32], we used PhageTerm [51] to predict the linear ends of packaged DNA. Native DNA termini are present once per linear DNA molecule, but non-native DNA ends produced during sequencing are distributed randomly along DNA molecules. Thus, reads that start at native DNA terminal positions occur more frequently than anywhere else in the genome. PhageTerm takes advantage of this to resolve DNA termini and predict phage packaging mechanisms [51]. PhageTerm identified the termini of packaged cp32 molecules at approximately 26 kb in a region lying immediately upstream of the *erp* loci (**Fig 6A**). In agreement with the PhageTerm results, when packaged reads were used to map the physical ends of packaged cp32 molecules, a sharp boundary in coverage depth is observed upstream of the *erp* loci in all cp32s (**Fig 6B–6F**). Notably, the intergenic region upstream of the *erp* loci is conserved across the cp32 isoforms found in diverse strains of Lyme disease spirochetes (**Fig 6G**) [15] and the linear cp32 ends identified by long-read sequencing converge at the same conserved terminal sequence motif (**Fig 6H**).

PhageTerm predicts that cp32s are packaged by a headful mechanism which supports the previously proposed headful genome packaging mechanism for cp32s [41]. Phages that use the headful packaging mechanism generate a concatemer containing several head-to-tail copies of their genome (**Fig 7A**). During headful packaging, a cut is made at a defined packaging site (*pac* site) and a headful (a little more than a full genome) of linear phage DNA is packaged. Once a headful is achieved, the phage genome is cut at non-defined sites, resulting in variable cut positions and size variation in packaged DNA, which we observe in packaged cp32 reads downstream of the initial cut site (**Fig 6B–6F**).

Our results suggest that the cp32 *pac* site is upstream of the *erp* loci. If the cp32 *pac* site is in this region, then DNA molecules containing the *pac* sequence are expected to be packaged into φBB-1 virions. To test this, we cloned the putative cp32-3 *pac* site (**Fig 6G**, black bar) into a derivative of the pBSV2 shuttle vector that lacks the promoter and MCS [52], transformed *B. burgdorferi* strain CA-11.2A, and induced lytic φBB-1 replication with 5% ethanol. Supernatants containing virions were collected, filtered, treated with chloroform, and DNase treated as described above. pBSV2 shuttle vector copy numbers were measured by qPCR using primers that target the pBSV2 kanamycin resistance (*kan*) cassette. To control for possible chromosomal DNA contamination, qPCR was also performed using primers targeting the chromosomal *flaB* gene. Final packaged pBSV2 copy numbers were calculated by subtracting *flaB* copy numbers from pBSV2 (*kan* cassette) copy numbers.

Copy numbers of packaged pBSV2 encoding the cp32-3 *pac* site were significantly ($p<0.001$) higher compared to virions collected from the supernatants of cells carrying an

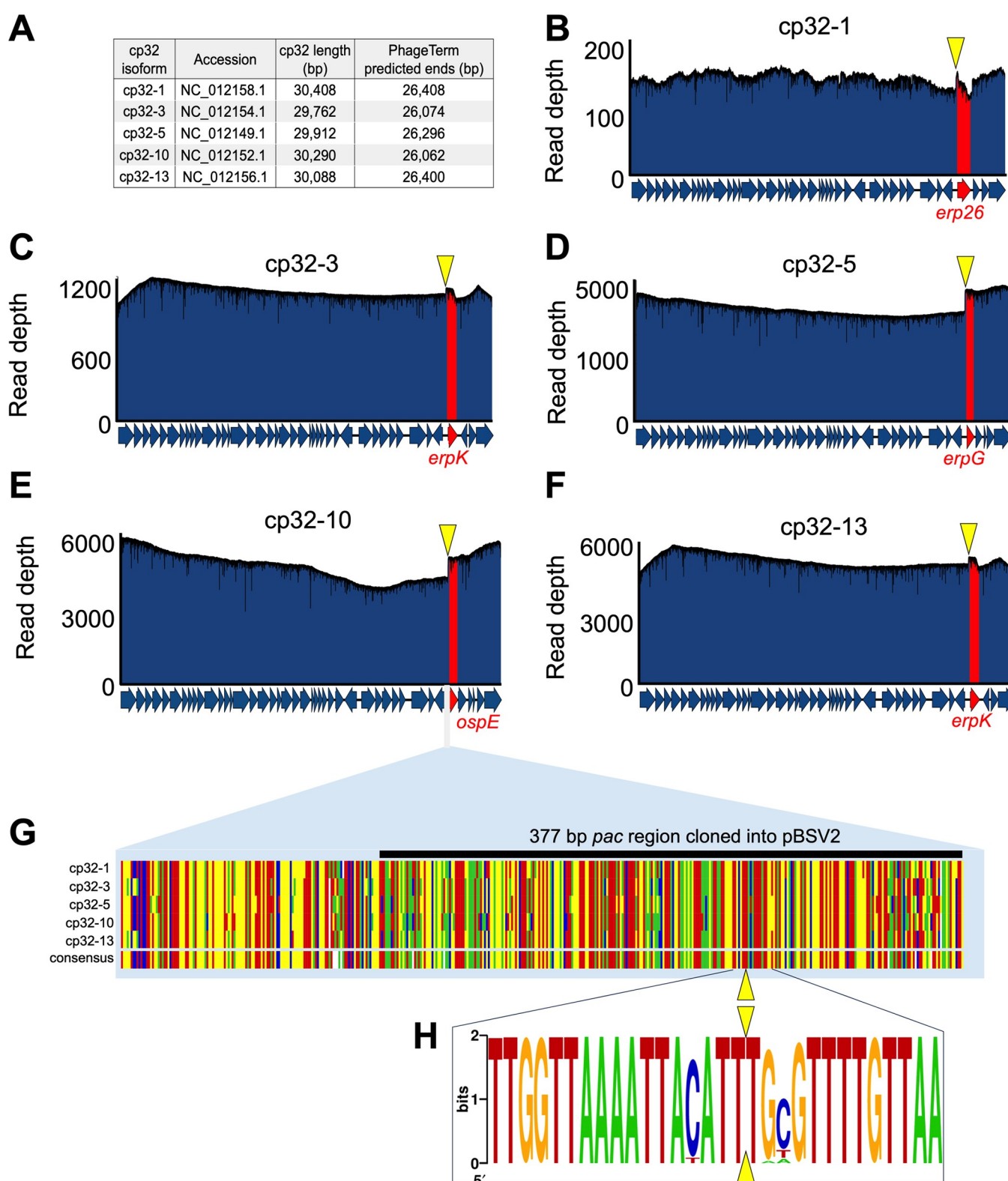

**Fig 6. cp32s are linearized upstream of the *erp* loci.** (A) PhageTerm was used to predict the linear ends of packaged cp32 molecules. (**B–F**) Nanopore reads were mapped to the indicated cp32s. Note the sharp boundary just upstream of the *erp* loci (highlighted in red). The yellow triangles indicate the PhageTerm predicted linear ends. (**G**) Alignments of the intergenic region upstream of the *erp* loci is shown for each cp32. Colors indicating A, T, C, or G are shown in panel H. The black line indicates the *pac* region that was cloned into a shuttle vector, as described in Fig 7. (**H**) A nucleic acid logo was constructed from 207 cp32 sequence alignments. Yellow triangles indicate the linear end of cp32 isoforms as predicted by PhageTerm and confirmed by long-read sequencing.

## A
### Headful packaging mechanism

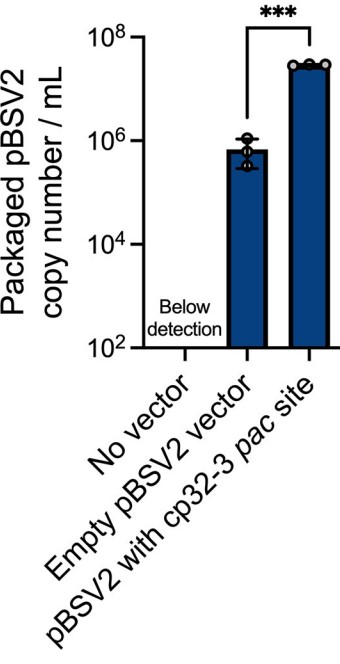

## B
### Packaging of shuttle vectors with the cp32 *pac* region into φBB-1 virions

**Fig 7. Shuttle vectors containing the cp32 *pac* region are preferentially packaged into φBB-1 virions.** (A) Schematic depicting the headful genome packaging mechanism. (B) After ethanol induction, φBB-1 virions were collected from CA-11.2A cells not carrying plasmid pBSV2 (No vector), cells transformed with empty pBSV2, or cells transformed with pBSV2 with the cp32-3 *pac* site (see Fig 6G for the cloned *pac* region). Copy numbers of pBSV2 packaged into φBB-1 virions were measured by qPCR. Data are the SE of the mean of three experiments, ***$p$<0.001.

empty pBSV2 vector (**Fig 7B**), indicating that DNA molecules that contain the *pac* site are preferentially packaged by φBB-1 virions.

Our results indicate that *B. burgdorferi* CA-11.2A produces more phage than strain B31 (**Fig 2A**), consistent with previous data [32]. This observation could be explained by mutations in the *pac* region or other regulatory elements that govern phage replication decisions. Indeed, comparing the *pac* region in the B31 cp32s to the CA-11.2A cp32s reveal numerous differences upstream of the cut site we identified (**S3A Fig**). Notably, the cut site itself is conserved across cp32s in B31 and CA-11.2A.

Phylogenetic analysis of the cp32 *pac* region in CA-11.2A and B31 reveal further insight. We find that the CA-11.2A cp32-1 isoform is packaged at a ~100-fold lower frequency than the other cp32 isoforms (see **Fig 5B and 5C**). Phylogenetic analysis reveals that the cp32-1 *pac* region is divergent from the other cp32 *pac* sites (**S3B Fig**). Similarly, prior observations in B31 by Wachter *et al.* [9] find that the cp32-9 isoform is not transcriptionally active compared to the other cp32 isoforms. Our phylogenetic analysis reveals that the *pac* region of cp32-9 is also divergent (**S3B Fig**). These results suggest that mutations in the *pac* region can affect DNA packaging efficiency into phage capsids.

### The cp32 prophages have conserved motifs that occur in a specific arrangement not found in other DNA sequences packaged by φBB-1 virions

To identify motif(s) that may be shared between the cp32s and other genomic elements that are packaged into φBB-1 virions (*e.g.*, lp54), we first used an iterative BLAST search to identify

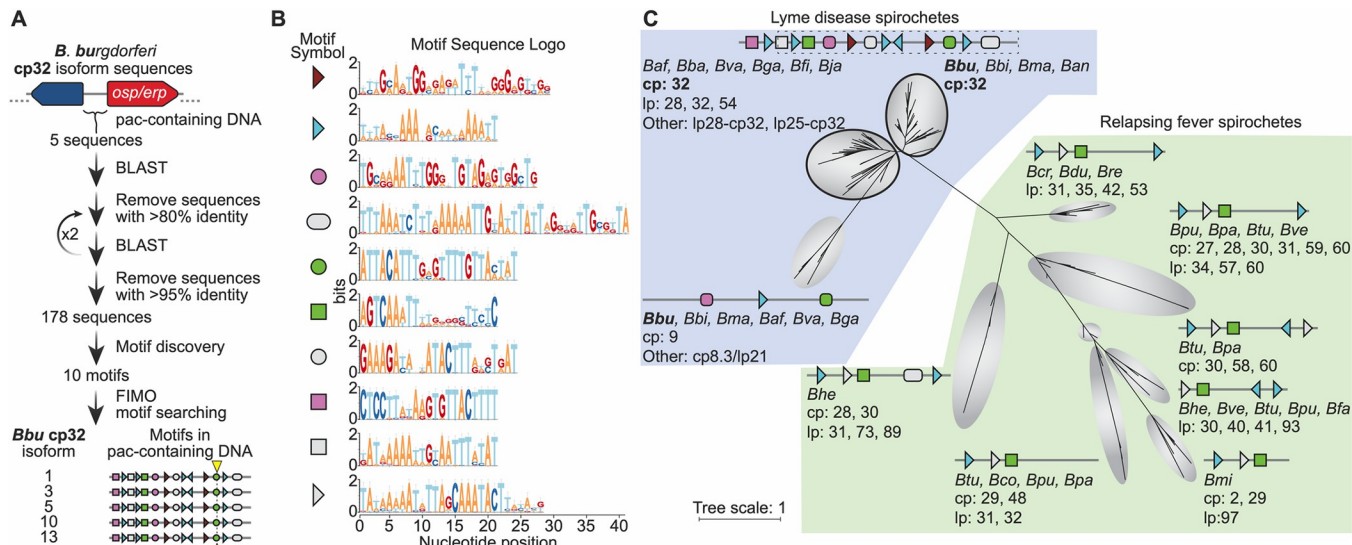

**Fig 8. Cp32 prophages have conserved motifs that occur in a specific arrangement around the *pac* site. (A)** Outline of bioinformatic strategy to identify motifs enriched in the *pac*-containing DNA sequence of cp32 isoforms. All *B. burgdorferi* cp32 isoforms have the same motifs in the *pac* region. The cp32 cut site is indicated by the yellow triangle. **(B)** Sequence logos of the motifs identified in panel A and schematized in panel C. Nine of the top ten motifs occur at least once in the *pac*-containing region of cp32 DNA sequences. Motifs represented with right or left facing triangles often occur as direct and/or indirect repeats. **(C)** Phylogenetic tree of non-redundant DNA sequences with homology to *B. burgdorferi* cp32 *pac*-region identified in panel A. For each clade, the bacterial species and type of plasmid are listed. For clarity in the figure, bacterial species names have been truncated to a three letter abbreviation consisting of the first letter of the genus and the first two letters of the species (*Borrelia afzelii*, *Baf*; *Borrelia andersonii*, *Ban*; *Borrelia bavariensis*, *Bba*; *Borrelia bissettiae*, *Bbi*; *Borrelia burgdorferi*, *Bbu*; *Borrelia coriaceae*, *Bco*; *Borrelia crocidurae*, *Bcr*; *Borrelia duttoni*, *Bdu*; *Borrelia fainii*, *Bfa*; *Borrelia finlandensis*, *Bfi*; *Borrelia garinii*, *Bga*; *Borrelia hermsii*, *Bhe*; *Borrelia japonica*, *Bja*; *Borrelia mayonii*, *Bma*; *Borrelia miyamotoi*, *Bmi*; *Borrelia parkeri*, *Bpa*; *Borrelia puertoricensis*, *Bpu*; *Borrelia recurrentis*, *Bre*; *Borrelia turicatae*, *Btu*; *Borrelia valaisiana*, *Bva*; *Borrelia venezuelensis*, *Bve*). There is variability in the motif architecture between sequences within a single clade; however, for clarity, a representative motif architecture discovered by MEME is shown [53]. The top two clades of sequences (outlined in black) are dominated by cp32 isoforms and the cp32 motif architecture, therefore a single motif scheme is shown for these two clades. The region of DNA and motifs cloned into the pBSV2 shuttle vector is outlined in dashes.

distantly homologous DNA sequences (**Fig 8**). A non-redundant list of these diverse DNA sequences were then used as an input dataset for sequence motif discovery via MEME [53]. All five cp32 isoforms found in *B. burgdorferi* CA-11.2A have the same specific arrangement of conserved sequence motifs around the *pac* region (**Fig 8A and 8B**) and these are conserved in cp32 isoforms across *B. burgdorferi* (**Fig 8C**). However, significant matches to these motifs were not identified in other CA-11.2A genetic elements packaged by ϕBB-1 (**S1 Data**), suggesting that packaging of non-cp32 DNA may require a pseudo-*pac* site that is smaller than the motifs identified or that packaging of non-cp32 DNA occurs spontaneously or through different mechanisms.

The complete or partial arrangement of motifs found around the *pac* site of *B. burgdorferi* cp32 isoforms is conserved in cp32 plasmids and some linear plasmids originating from other Lyme and relapsing fever *Borrelia* (21 species total) (**Fig 8C**). The iterative BLAST search also revealed that a diverse set of circular and linear plasmids in a broader set of *Borrelia* species share some of the motifs found in *B. burgdorferi* cp32 isoforms. In total, linear or circular plasmid sequences from 21 different *Borrelia* species (both Lyme disease and relapsing fever spirochetes) had homology to the *B. burgdorferi* cp32 *pac*-containing DNA sequences (**Fig 8C**).

## Deciphering the structure of linear plasmids packaged by ϕBB-1

After the cp32s, lp54 is a major DNA species packaged by ϕBB-1 (**Fig 5C**). Lp54 is a linear plasmid with covalently closed telomeres that is present in all Lyme disease *Borrelia* with about a third of its encoded genes being paralogues to genes encoded on the cp32s [6,54]. *De novo*

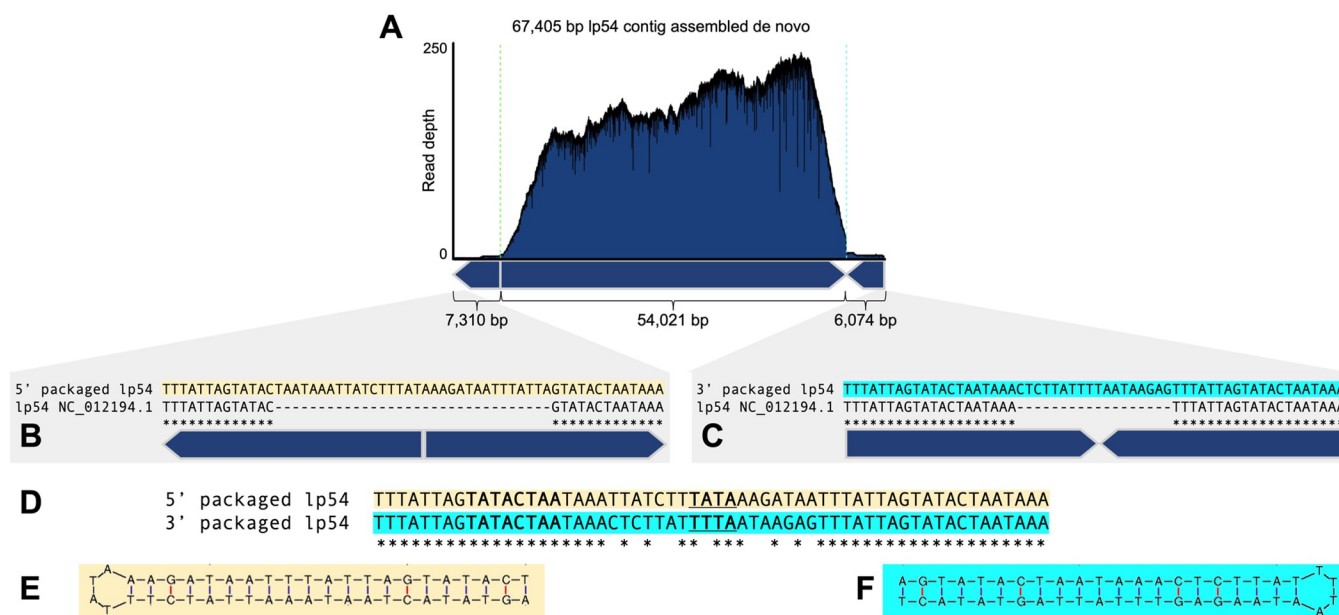

**Fig 9. Full-length lp54 with fully resolved telomeres are recovered from ϕBB-1-packaged DNA. (A)** *De novo* assembly of packaged reads produced a 67,405-bp contig with tail-to-tail and head-to-head junctions. **(B and C)** Sequences at the packaged 5′ junction (light orange) or the 3′ junction (cyan) are compared to the lp54 reference sequence NC_012194.1. **(D)** Alignments of the tail-to-tail and head-to-head junctions reveals a variable 18-bp sequence in between the conserved inverted repeats. **(E and F)** Predicted hairpin structures are shown for each end of lp54. The loop sequence for each hairpin is underlined in panel D.

assembly of packaged lp54 reads produces a 67.4 kb contig consisting of full-length lp54 (54,021 bp, NC_012194.1) flanked by sequences containing tail-to-tail (7,310 bp) and head-to-head (6,074 bp) junctions (**Fig 9A**). Read depth for lp54 was >100 for most of the contig; however, read depth drops precipitously at both tail-to-tail and head-to-head junctions (**Fig 9A**), suggesting that the telomeres of lp54 interfere with sequencing.

*B. burgdorferi* telomeres contain inverted repeat sequences [55] and we identified the CA-11.2A lp54 inverted repeat sequence as 5′–TTTATTAGTATACTAATAAA (**Fig 9B and 9C**, boxed sequences). Our sequencing of the telomeric ends of lp54 extends the reference sequence at the left telomeric end by seven nucleotides (**Fig 9B,** underlined). Further, compared to the lp54 reference sequence, the packaged left and right junction-spanning sequences each encode an additional 18 bp of sequence (**Fig 9B and 9C**). These sequences, although unique at each end (**Fig 9D**), form perfect hairpin structures (**Fig 9E and 9F**). Overall, these data suggest that lp54 molecules with complete telomere sequences are packaged into virions. However, whether linear lp54 with covalently closed telomeres or lp54 replication intermediates that contain head-to-head and tail-to-tail junctions are packaged is unclear.

The *de novo* assembly approach applied to lp54 was also successful in resolving the telomeric ends of other linear elements of the CA-11.2A genome, including the linear chromosome and plasmids lp17, lp56, and lp38 (**Fig 10**). Conserved elements for each telomere are highlighted [56–61]. Additionally, we were able to resolve left and right telomeres for lp36 (**Fig 10**), providing yet further evidence that lp36 is not fused to lp28-4.

## Discussion

In nature, Lyme disease spirochetes exist as diverse populations of closely related bacteria that possess sufficient antigenic variability to allow them to co-infect and reinfect non-naïve

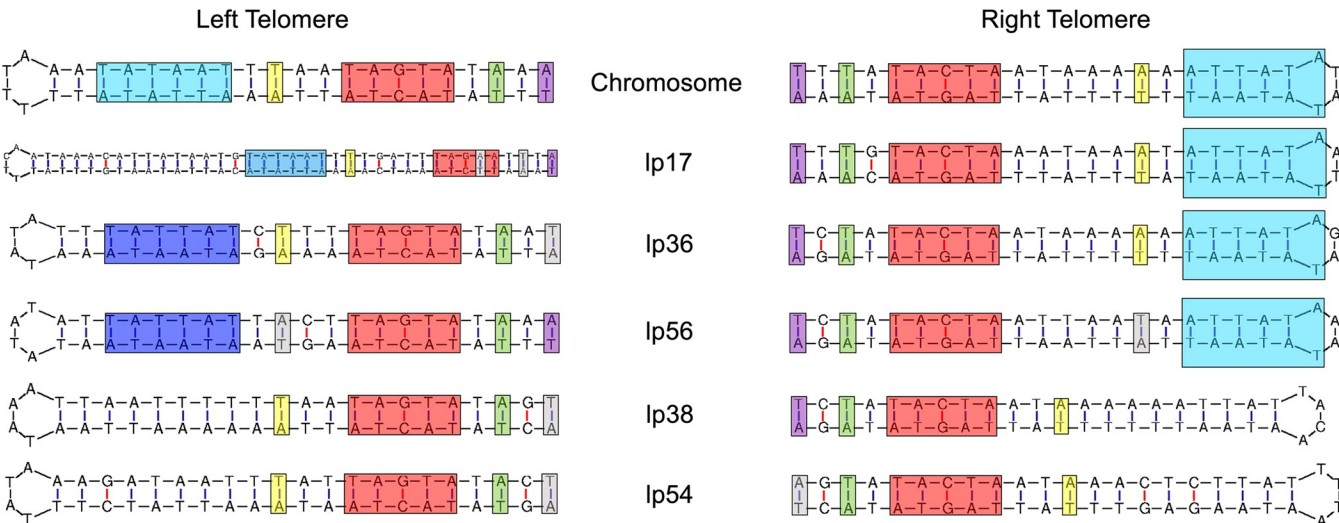

**Fig 10. Packaged reads resolve the telomeric ends of the linear chromosome and most linear plasmids in the CA-11.2A genome.** Reads spanning tail-to-tail or head-to-head junctions of the linear chromosome or the indicated linear plasmids form perfect hairpin structures. Conserved regulatory elements for each telomere are highlighted.

vertebrate hosts [62–73]. Moreover, horizontal gene transfer between Lyme disease spirochetes has been extensively documented [19,74–78]. Nevertheless, the mechanism underlying horizontal genetic exchange among Lyme disease spirochetes has remained undefined. Our study implicates φBB-1 in mediating horizontal gene transfer between Lyme disease spirochetes.

Horizontal gene transfer between heterologous spirochetes likely occurs in the tick midgut during and immediately after a blood meal when spirochete replication rates and densities are at their highest. φBB-1 replication is also induced in the tick midgut during a bloodmeal [9,29,30] with implications for their facilitation of horizontal gene transfer evidenced by homologous recombination between cp32 isoforms [15–17] and the horizontal transfer of cp32s between *Borrelia* strains [21]. These observations suggest a conserved phage receptor is present across the genospecies; however, the identity of the φBB-1 is not yet known.

Our sequencing data suggest that φBB-1 virions package large portions of the *B. burgdorferi* genome, giving φBB-1 the potential to mobilize numerous beneficial alleles during the enzootic cycle via generalized transduction. For example, the circular cp32 prophages are highly conserved across the *Borrelia* genus [26]; however, cp32 isoforms contain variable regions that encode outer membrane lipoproteins such as Mlp and OspE/OspF/Elp, which are known to facilitate the *B. burgdorferi* lifecycle [24,26,27,79]. The linear plasmid lp54 encodes the outer membrane lipoproteins OspA and OspB, which are required for *B. burgdorferi* to colonize the tick midgut [80–82]. The outer membrane lipoprotein OspC, which is required for *B. burgdorferi* to infect a vertebrate host, is encoded by the circular plasmid cp26 [62,77,83]. These alleles (and many others) are packaged by φBB-1, which is consistent with a role for phage-mediated transduction of genes encoding essential membrane lipoproteins between heterologous spirochetes.

In *B. burgdorferi*, the linear chromosome is highly conserved as are the circular plasmids cp32 and cp26 and the linear plasmids lp17, lp38, lp54, and lp56 are all evolutionarily stable [4–6,16,84]. However, other plasmids distributed across the genospecies show considerably more variation, encode mostly (87%) pseudogenes, and are thought to be in a state of evolutionary decay [6]. The packaged plasmids for which we recovered full-length contigs include the cp32s, cp26, lp17, lp38, lp54, and lp56—the same plasmids that are evolutionarily stable

across the genospecies [4–6,16,84]. These observations suggest that genes encoded on φBB-1-packaged plasmids are under positive selection, possibly due to the continuous transduction between Lyme disease spirochetes during the enzootic cycle.

In addition to providing evidence that φBB-1 virions package large portions of the *B. burgdorferi* genome, our study provides insight into φBB-1 virion structure and identifies virion proteins present in φBB-1. Using mass spectrometry-based proteomics, we confirm that putative capsid and structural genes encoded by the cp32s, such as the major capsid protein P06, are indeed translated and assembled into mature φBB-1 virions.

Our long-read sequencing studies indicate that φBB-1 packages full-length linear cp32 molecules via a headful mechanism using *pac* sites. The headful packaging mechanism is used by numerous phages and was first described for *E. coli* phage T4 in 1967 [85]. After injecting linear DNA into a new host, the phage genome re-circularizes before continuing its replication cycle. Genes encoded near the ends of linear phage genomes are subject to copy number variation and recombination as the phage genome re-circularizes [86]. Our data suggest that the conversion of linear cp32 molecules into circular cp32 molecules occurs in the vicinity of the *erp* locus, which would facilitate recombination with polymorphic *erp* alleles encoded by other cp32 isoforms in diverse *B. burgdorferi* hosts.

In this study, the packaging of specific cp32 isoforms was biased: cp32-3, cp32-5, cp32-10, and cp32-13 were predominantly packaged while cp32-1 was rarely packaged. This result is consistent with observations by Wachter *et al.* where cp32 isoform copy number and transcriptional activity were not uniform across all cp32 isoforms in *B. burgdorferi* strain B31: cp32-1, cp32-3, and cp32-6 were predominantly induced (highest copy numbers) and had the highest transcriptional activity while cp32-9 was not induced and was transcriptionally inactive [9]. Variability in the *pac* region or other regulatory elements involved in cp32 induction may explain why different cp32 isoforms replicate and/or are packaged at different rates. Pseudo-*pac* site homology to bona fide *pac* sites affects generalized transduction frequencies by phages like P22 [87–89]. However, generalized transduction is unpredictable and any part of the bacterial genome is likely packaged at low frequencies [90]. The motifs that are found most broadly in the *pac* region (*e.g.*, **Fig 8C**, blue triangle and green square) may represent pseudo-*pac* sites for conserved host factors that are present in all *Borrelia* species whereas the other motifs may represent protein-binding sites or regulatory sequences that are specific to given prophage or plasmids.

In the intergenic region upstream of the *erp* loci, we identified a 377-bp region that contains the cp32 *pac* signal. Introducing the cp32 *pac* region to a shuttle vector facilitated the packaging of the shuttle vector into φBB-1 virions. Our identification of the cp32 *pac* site will be useful for the engineering of recombinant DNA that can be packaged into virions that infect spirochetes, giving φBB-1 the potential for use as a tool for the genetic dissection and manipulation of Lyme disease spirochetes.

After the cp32s, lp54 was the most frequently packaged plasmid. This may be related to the evolutionary origins of lp54: about one-third of the genes encoded by lp54 are paralogous to cp32-encoded genes and lp54 is thought to have emerged from an ancient recombination event between a cp32 and a linear plasmid [6]. In addition, lp54 encodes putative phage proteins including a porin (BBA74) [91] and phage capsid proteins that are highly conserved across the genospecies [92], which we detected in purified virions by mass spectrometry. While we observed virions with a distinct elongated capsid morphology, virions with a notably smaller capsid morphology have been observed after induction *in vitro* [9,31,32]. These observations raise the possibility that lp54 may be a prophage, although it is not clear if lp54 produces its own capsids, relies on cp32-encoded capsids, or if both lp54 and cp32 capsid proteins assemble to produce chimeric virions.

Our long-read dataset contained reads that spanned head-to-head and tail-to-tail junctions in lp54. These reads allowed us to define the lp54 telomere sequences; however, whether full-

length lp54 molecules are packaged or at which stage of the replication cycle lp54 is packaged is unknown. In *B. burgdorferi*, both the linear chromosome and linear plasmids have covalently closed hairpin telomeres and replicate via a telomere resolution mechanism [57,59,93,94]. Examination of a naturally occurring lp54 dimer in *B. valaisiana* isolate VS116 suggests that a circular head-to-head dimer is produced during lp54 replication prior to telomere resolution and replication completion [95]. Linear, covalently closed lp54 molecules may be packaged or lp54 replication intermediates may be packaged.

As obligate vector-borne bacteria, Lyme disease spirochetes live relatively restrictive lifestyles that might be expected to i) limit their exposure to novel gene pools, ii) enhance reductive evolution, and iii) favor the loss of mobile DNA elements. A role for ϕBB-1 in mediating the transduction of beneficial alleles between heterologous spirochetes in local vector and reservoir host populations may explain why cp32 prophages are ubiquitous not only amongst Lyme disease spirochetes, but also relapsing fever spirochetes.

## Methods

### ϕBB-1 induction

*Borrelia burgdorferi* B31 or CA-11.2A was grown in BSK-II growth medium to $7 \times 10^7$/mL and centrifuged at $6,000 \times g$, 10 min, 35°C to pellet cells, which were resuspended in fresh media to a density of $2 \times 10^8$/mL. EtOH was added to a final concentration of 5% and the resuspended culture was incubated at 35°C for an additional 2 hours to induce phage production. The induced culture was then centrifuged at $6,000 \times g$, 10 min, 35°C and the pellet was resuspended in fresh media to a density of $5 \times 10^7$/mL after which it was incubated at 35°C for 72 hours to produce phage. After 72 hours, the culture was centrifuged at $6,000 \times g$ for 10 min to remove cells and the phage-containing supernatant was filtered twice through 0.2 μm filters before storage at 4°C.

### cp32 qPCR

For qPCR, 100 μL of filtered culture supernatant was mixed with 20ul of chloroform to eliminate remaining intact cells and then centrifuged to separate the phases. 80 μL of the aqueous phase was transferred to a new tube, mixed with 0.8 μL of 100X DNase I reaction buffer (1M Tris-HCl pH 7.5, 250 mM $MgCl_2$, 50 mM $CaCl_2$) and DNase treated with 0.8U DNase I for 1 hour at 37°C. Following DNase treatment, supernatants were mixed with 20 μl chloroform to inactivate DNase, spun to separate phases and the aqueous phase added directly to a qPCR reaction (0.5 μL treated supernatant/10 μL total reaction volume). qPCR was performed using SsoAdvanced Universal Inhibitor-Tolerant SYBR green supermix (BioRad, Hercules, CA) following maufacturer's instructions, primers that target a conserved cp32 intergenic region between *bbp08* and *bbp09* (5′–CTTTACACATATCAAGACCTTAAC, 5′– CAAACCACCCAATTTCCAATTCC) and the *flaB* gene to control for *B. burgdorferi* chromosomal DNA contamination (5′–TCTTTTCTCTGGTG AGGGAGCT, 5′– TCCTTCCTGTTGAACACCCTCT) [96] at an empirically determined annealing temperature of 55°C. Absolute cp32 and *flaB* copy numbers were calculated from a standard curve generated using a cloned copy of the target sequences. To estimate phage number for CA-11.2A and correct for any remaining unpackaged cp32 plasmids, five times the number of detected *flaB* copies was subtracted from the absolute cp32 starting quantity.

### ϕBB-1 virion purification for DNA extraction

Centrifuged, filtered phage supernatants were treated with 1/10th volume of chloroform to lyse any remaining cells and chloroform was allowed to separate at 4°C overnight. The aqueous layer was transferred to a new vessel and mixed with saturated ammonium sulfate to a final

concentration of 50%. NaOH was slowly added during ammonium sulfate addition to maintain pH based on the BSK-II phenol red indicator and the final pH was adjusted to 7.5. Precipitations were incubated overnight 4˚C and then centrifuged at $10,000 \times g$ for 30 minutes (4˚C) to collect phage pellets. Precipitated phages were gently resuspended in SM buffer (100 mM NaCl, 8 mM $MgSO_4$, 50 mM Tris-HCl, pH 7.5) overnight at 4˚C.

### φBB-1 electron microscopy imaging

Purified virions (3–4 μL) were absorbed to the surface of freshly glow-discharged, formvar-coated 200 mesh copper grids and negatively stained with 5 μl of 2% methylamine vanadate (Nanoprobes, Yaphank, NY) prior to viewing on a Hitachi HT7700 transmission electron microscope (Hitachi-High-Technologies Corporation, Tokyo, Japan).

### φBB-1 virion proteomics

Purified virions (200 μg total protein) were reduced, alkylated, and purified by chloroform/methanol extraction prior to digestion with sequencing grade modified porcine trypsin (Promega). Peptides were separated on an Acquity BEH C18 column (100 x 1.0 mm, Waters) using an UltiMate 3000 UHPLC system (Thermo). Peptides were eluted by a 50 min gradient from 99:1 to 60:40 buffer A:B ratio (Buffer A = 0.1% formic acid, 0.5% acetonitrile. Buffer B = 0.1% formic acid, 99.9% acetonitrile). Eluted peptides were ionized by electrospray (2.4 kV) followed by mass spectrometric analysis on an Orbitrap Eclipse Tribrid mass spectrometer (Thermo) using multi-notch MS3 parameters. MS data were acquired using the FTMS analyzer over a range of 375 to 1500 m/z. Up to 10 MS/MS precursors were selected for HCD activation with normalized collision energy of 65 kV, followed by acquisition of MS3 reporter ion data using the FTMS analyzer over a range of 100–500 m/z. Proteins were identified and quantified using Mascot [97] with a parent ion tolerance of 2.5 ppm and a fragment ion tolerance of 0.5 Da. Peptides with a probability score $[-10Log_{10}(P)] > 70$ were considered significant.

### Packaged φBB-1 DNA purification

For DNA extractions, phage were collected and precipitated as described above, with the addition of a DNase treatment prior to ammonium sulfate precipitation. The aqueous phage of chloroform supernatants were mixed with 1/100th volume 100X DNAse buffer and 1U/mL DNase I followed by incubated at 37˚C for 3 hours and by 4˚C overnight. For samples subjected to population sequencing, high molecular-weight salmon sperm DNA (1.7 μg/mL, a concentration that approximates the amount of DNA released by $3 \times 10^8$ lysed bacterial cells per milliliter of media) was added prior to DNase digestion to assess carryover of DNA contained outside of phage capsids.

After ammonium sulfate precipitation and resuspension of phage pellets in SM buffer, EDTA was added to a final concentration of 5 mM and SDS to a final concentration of 0.5%. After addition of 20 μg/mL RNase and incubation at room temperature for 20 minutes, phage capsids were digested with 200ug/mL proteinase K at 55˚C for 1 hour. Samples were extracted twice with an equal volume of phenol-chloroform-isoamyl alcohol (25:24:1) followed by a single extraction with an equal volume of chloroform-isoamyl alcohol (24:1) using Qiagen Maxtract High Density medium (Qiagen, Hilden, Germany). NaCl was added to 300 mM and DNA was precipitated with 2.5 volumes of 100% EtOH at -20˚C overnight. DNA was pelleted by centrifugation ($14,000 \times g$ for 20 min at 4˚C), washed 3X with 70% EtOH and re-spun for 20 min, at $14,000 \times g$ 4˚C. The DNA pellet was gently air-dried followed by resuspension in 10mM Tris-HCl, pH 8.5 at 4˚C overnight.

## Nanopore sequencing

Sequencing libraries were prepared according to manufacturer's instructions using library kit SQK-LSK112, native barcoding kit SQK-NBD112.24 and 500 ng of purified phage DNA (Oxford Nanopore, Oxford, UK). Libraries were sequenced on a MinION MK1-B using a FLO-MIN112 flowcell and default settings until pores were exhausted. Basecalling and demultiplexing was performed with Guppy 6.4.6 using the super high accuracy (SUP) model (dna_r10.4_e8.1_sup.cfg) and default parameters. Run quality control measures were checked with MinIONQC (v1.4.1) [98] and FastQC (v0.11.9). Adaptor trimming was performed using s (v0.2.4) [99]. Reads were deposited in the NCBI BioProject database accession PRJNA1059007 and in S2 Data.

## Sequence analysis pipeline

Adapter-trimmed long-reads with quality scores $\geq 7$ were used to isolate $\geq$ 5kb reads using Filtlong (v0.2.1). $\geq$5kb reads were mapped to the reference *B. burgdorferi* CA-11.2A genome (RefSeq assembly: GCF_000172315.2) with minimap2 (v2.26-r1175) [100]. Primary mapping reads with MAPQ >20 were isolated by contig, filtered, and converted to final file formats using Samtools (v1.17) [101] and SeqKit (v2.5.1) [102]. Read statistics for each replicate were graphed and viewed using GraphPad Prism (v10.1.1). For each contig, *de novo* assemblies were created using Trycycler (v0.5.4) [103], which relied on input assemblies from Flye (v2.9.2-b1786) [104], Raven (v1.8.3) [105], and Minimap2/Miniasm/Minipolish (v2.26-r1175/v0.3-r179/v0.1.2) [100,106]. The long-read *de novo* assemblies were then polished with short reads using Minipolish (v0.1.2) [106]. The telomeres of the linear chromosome and linear plasmids were manually identified in SnapGene (v5.3.3), and the hairpin structures were predicted by the Mfold webserver (http://www.unafold.org/mfold/applications/dna-folding-form.php) [107]. The terminal ends of the cp32 prophage genomes were predicted using PhageTerm through the Galaxy webserver (https://galaxy.pasteur.fr/) [51], via input of the $\geq$5kb long-read sequences. Coverage maps of the primary mapping or primary and supplementary mapping reads were created by mapping $\geq$ 5kb long-reads to the *de novo* assembled CA-11.2A genome or the reference *B. burgdorferi* CA-11.2A genome with Minimap2, converted to final file formats using Samtools, and viewed using R (v4.3.2) and ggplot2 (v3.4.4).

## *Pac* site cloning and qPCR

The putative *pac* region from CA-11.2A genomic DNA was amplified using primers 5′-TAGA CATGAGCGGCCGCAAGACAAGCTCCTTATAAGTGTTACT-3′ and 5′-ATAGCTAGAT GCGGCCGCTTACTCCGTAACTCTAAAGAATAATGC-3′, purified and digested with NotI and cloned into Not-I-digested pBSV2_2 [52] to create a shuttle vector in which the CA-11.2A *pac* region is maintained but cannot be expressed. Vector sequences were verified using long-read sequencing and transformed into CA-11.2A via electroporation [108]. Clones were PCR-screened for maintenance of resident plasmids as previously described using published primers for *B. burgdorferi* cp32-1, cp26, cp32-3 (which target CA-11.2A cp32-5), cp32-6 (which target CA-11.2A cp32-3), lp28-3, lp17, lp54, lp28-4 [109] and CA-11.2A-specific primers for cp32-3 (5′-TGGGTTGTAGAGTGGCTGTG-3′, 5′-TCACCACTTGCGTAATTCTTGC-3′), cp32-10 (5′-TAGAGCAAAGTCTTGAAAAGACAAC-3′, 5′-CCCACGCTTTGTTGAGACC-3′) and cp32-13 (5′- AATCTGGGCTGTAGAGCAGG-3′, 5′-CTGCTCCTGAGGCTCATCC-3′). Clones transformed with *pac* plasmids or the empty vector were grown in triplicate to late-log phase in BSK-II and used to generate phage as described above. Encapsidated vector was measured directly from DNase-treated culture supernatants as described above using qPCR primers that target the *kan* resistance gene on pBSV2_2 (5′-CACCGGATTCAGTCGTCACT-

3′, 5′-GATCCTGGTATCGGTCTGCG-3′, 120 bp product). A cloned copy of the *kan* PCR product was used to generate a standard curve for absolute quantification.

## Identification of conserved motifs in *B. burgdorferi* cp32 isoforms

The roughly 430 nucleotides upstream of the *erp26*, *erpK*, *erpG*, *ospE* and *erpK* genes of the *B. burgdorferi* CA-11.2A cp32 isoforms cp32-1, cp32-3, cp32-5, cp32-10 and cp32-13 respectively were used as queries for a discontinuous MegaBLAST against the NCBI Nucleotide collection database. The results from these first five BLASTs were combined and sequence hits with more than 80% identity were removed with CD-HIT [110]. The resulting representative sequences were used as queries for discontinuous MegaBLAST against the NCBI Nucleotide collection database, and sequence hits with more than 80% identity were removed with CD-HIT [110]. This process was iterated twice more for a total of three MegaBLAST searches with a representative list of 80% identity query sequences. The sequence hits from the final MegaBLASTs were combined and sequences with more than 95% identity were removed with CD-HIT [110], generating a list of 178 sequences. These 178 sequences were used as an input dataset for the MEME webserver [53], with custom parameters of "Maximum Number of Motifs" set to "10", and "Motif Site Distribution" set to "Any number of sites per sequence". MEME identified motifs in 160 of the input sequences. The Position Weight Matrices (PWMs) of the 10 motifs identified by MEME were used as inputs for FIMO [111] to search for significant sequence matches (q-value < 0.001) in the *B. burgdorferi* chromosome and the *B. burgdorferi* cp32-1, cp32-3, cp32-5, cp32-10, cp32-13, cp26, lp17, lp54 plasmid DNA sequences. The cp32 isoforms had nine highly conserved sequence motifs, some motifs present in multiple copies and arranged in a conserved architecture. The cp26, lp17, lp54 and chromosome sequences did not contain this conserved architecture of nine motifs (see S1 Data). The sequence logo of each motif was generated by taking the sequence fragments that MEME used to make each PWM, and submitting these sequence fragments to the WebLogo 3.0 webserver [112]. The iterative discontinuous MegaBLAST searches had introduced eukaryotic sequence fragments into the list of 178 non-redundant sequences, suggesting that the search likely reached an endpoint and found most of the related sequences in the NCBI database. To generate a phylogenetic tree, eukaryotic sequence fragments were first removed, and the remaining 149 non-redundant sequences were aligned using the MAFFT webserver [113], with custom parameters of "Direction of nucleotide sequences" set to "Adjust direction according to the first sequence", and "Strategy" set to E-INS-2. The resulting alignment was used as input for the IQ-TREE webserver [114,115], with the following command-line: path_to_iqtree -s *.fasta -st DNA -m TEST -bb 1000 -alrt 1000. TreeViewer was used to display the phylogenetic tree [116].

## Supporting information

**S1 Fig. Packaged read depth across the *de novo* CA-11.2A genome assembly.** *De novo* assembly of packaged reads >5kb produced the indicated contigs. Read coverage was then mapped to each contig. **(A–H)** Read coverage across the CA-11.2A chromosome or indicated plasmids are shown. Coverage maps for the cp32s and lp54 are shown in Figs 6 and 9, respectively.
(TIFF)

**S2 Fig. Whole genome sequencing of the CA-11.2A genome reveals that plasmid lp36/lp28-4 resolves into two separate episomes. (A)** The CA-11.2A genome was sequenced using long-read technology. Reads were aligned to the lp36/lp28-4 reference sequence (NC_012202.1) and read depth plotted. **(B)** Schematic of PCR design. Primers 1 and 2 flank

the lp36/lp28-4 junction, with primer 1 annealing to lp36 and primer 2 annealing to lp28-4, creating a 620 bp product if joined. Primers 3 and 4 anneal to lp36 DNA, creating an 813 bp product if present. Primers 5 and 6 anneal to lp28-4 DNA, creating a 1,115-bp product if present. **(C)** The presence or absence of lp36, lp28-4, or lp36/lp28-4 was confirmed by PCR.
(TIFF)

**S3 Fig. *B. burgdorferi* CA-11.2A and B31 *pac* region comparison. (A)** The pac region from the cp32s in *B. burgdorferi* B31 and CA-11.2A were aligned by ClustalW. The cut site is indicated by yellow arrows. **(B)** Phylogenetic analysis of the *pac* regions was performed by constructing an uncorrected neighbor joining tree.
(TIFF)

**S1 Table. Bacteriophage proteomics significant hits.**
(XLSX)

**S2 Table. Maximum read lengths recovered for each element in the CA-11.2A genome**
(TIFF)

**S1 Data. Data set used to identify DNA motifs in the cp32 pac region.**
(ZIP)

**S2 Data. DNA sequence dataset.**
(ZIP)

## Acknowledgments

We are grateful to Patti Rosa for helpful discussions and to the IDeA National Resource for Quantitative Proteomics Center at the University of Arkansas for their assistance with proteomic analyses of phage virions.

## Author Contributions

**Conceptualization:** D. Scott Samuels, Patrick R. Secor.

**Data curation:** Margie Kinnersley, Patrick R. Secor.

**Formal analysis:** Dominick R. Faith, Margie Kinnersley, Andrew Santiago-Frangos, Patrick R. Secor.

**Funding acquisition:** Patrick R. Secor.

**Investigation:** Dominick R. Faith, Margie Kinnersley, Diane M. Brooks, Dan Drecktrah, Eric Luo, Andrew Santiago-Frangos, Jenny Wachter, D. Scott Samuels, Patrick R. Secor.

**Methodology:** Dominick R. Faith, Margie Kinnersley, Diane M. Brooks, Laura S. Hall, Eric Luo, Andrew Santiago-Frangos, Jenny Wachter, Patrick R. Secor.

**Project administration:** Patrick R. Secor.

**Supervision:** D. Scott Samuels, Patrick R. Secor.

**Visualization:** Patrick R. Secor.

**Writing – original draft:** Dominick R. Faith, Patrick R. Secor.

**Writing – review & editing:** Dominick R. Faith, Margie Kinnersley, Diane M. Brooks, Dan Drecktrah, Laura S. Hall, Eric Luo, Andrew Santiago-Frangos, Jenny Wachter, D. Scott Samuels, Patrick R. Secor.

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
