## [Decision Letter · Decision Letter 0]

14 Feb 2024

Dear Dr. Secor,

Thank you very much for submitting your manuscript "Characterization and genomic analysis of the Lyme disease spirochete bacteriophage ϕBB-1" for consideration at PLOS Pathogens. As with all papers reviewed by the journal, your manuscript was reviewed by members of the editorial board and by several independent reviewers. The reviewers appreciated the attention to an important topic. Based on the reviews, we are likely to accept this manuscript for publication, providing that you modify the manuscript according to the review recommendations.

Sincerely,

John M Leong

Pearls Editor

PLOS Pathogens

David Skurnik

Section Editor

PLOS Pathogens

Michael Malim

Editor-in-Chief

PLOS Pathogens

orcid.org/0000-0002-7699-2064

Reviewer Comments (if any, and for reference):

Reviewer's Responses to Questions

**Part I - Summary**

Reviewer #1: This manuscript provides a detailed characterization and genomic analysis of the conserved temperate bacteriophage ϕBB-1 of Borrelia burgdorferi. The study reveals that ϕBB-1 packages various genetic elements, including linear cp32s, plasmids, and fragments of the linear chromosome for phage transduction. It is found that ϕBB-1 packages linear cp32s via a headful mechanism. The authors identify the cp32 pac region and show that plasmids containing this region are preferentially packaged into ϕBB-1 virions. This opens up the possibility of using this phage for the genetic engineering of B. burgdorferi. By sequencing ϕBB-1 packaged DNA, the authors provided insight into the genetic material packaged by this phage, which is relevant to understanding the biology of the phage and its role in horizontal gene transfer between B. burgdorferi strains.

Reviewer #2: The manuscript by Faith et al., details the robust characterization of the B. burgdorferi phage ɸBB-1 produced by the prophages encoded by the cp32 plasmids. Mass spectrometry analysis of purified ɸBB-1 identified capsid and structural proteins encoded by the cp32 plasmids. Using the Nanopore MinION platform the team determined the B. burgdorferi DNA packaged by ɸBB-1. Removal of unpackage DNA was well controlled for by tracking the removal of spiked salmon-sperm DNA. As expected, cp32 DNA comprised the majority of the packaged reads. However, the packaged reads also included DNA located on the chromosome, lp54 and other plasmids. Most significantly this work identified the pac site necessary and sufficient for ɸBB-1 packaging of DNA. In addition, sequence analysis of the ɸBB-1 packaged DNA allowed for resolution of the telomeric ends of the packaged linear plasmids. In all, this manuscript describes a strong body of work. The findings provide important insight into the function and capabilities of ɸBB-1, which have potential implications for evolution of the B. burgdorferi genome as well as novel molecular genetic tools.

Reviewer #3: The Borrelia burgdorferi (Bb) bacteriophage φBB-1 was first described over twenty years ago. Over the ensuing years, studies by Samuels, Eggers and colleagues have provided additional incremental information demonstrating that φBB-1 can transduce cp32 plasmid DNA into a recipient strain, establishing cp32s as putative prophage. The demonstration that φBB-1 could transduce other DNAs suggests the possibility that it can mediate horizontal gene transfer among Bb leading to strain diversity. In this manuscript, the authors describe the use of long-read sequencing to identify the DNAs packaged by φBB-1, identify the cp32 pac site and the mechanism for packaging. This study is a tour-de-force providing a major step forward for understanding the biology of Bb bacteriophage and their potential role in mediating Bb strain diversity and use for Bb genetic manipulation. The manuscript is masterfully written. Some of the experimental approaches are quite complex, but the authors’ use of well-drawn schematic diagrams and clearly written legends make the experimental designs understandable to the non-expert reader. The methods and approaches are all appropriate and the results are lucidly presented.

**Part II – Major Issues: Key Experiments Required for Acceptance**

Reviewer #1: 1. The authors claim that ϕBB-1 is a generalized transducing phage, but this seems based solely on finding low coverage reads of the linear chromosome with no obvious pac sites. Because generalized transduction wasn’t shown in this study, more evidence is needed to substantiate this claim. What was the size distribution of chromosomal reads? If there were a peak read length matching the headful packaging size of ϕBB-1, then this would be good evidence of generalized transduction.

2. It should be mentioned what the receptor for ϕBB-1 is. Is it known? And, if so, is it highly conserved across the genospecies?

3. Can you get viable ϕBB-1 in an lp54 minus strain, or is Ip54 essential?

Reviewer #2: (No Response)

Reviewer #3: None noted.

**Part III – Minor Issues: Editorial and Data Presentation Modifications**

Reviewer #1: 1. Figures 2A and 7B: The y-axis scales are not linear, and there is no way to interpret the data near the bottom. For example, the Figure 2A y-axis goes from 0 to 1 x 108, then to 2 x 108, etc., so there is no way to know what the copy number for B31 is. Replace the zeros with the correct numbers. The other graphs in the manuscript did it right.

2. Figure 8B does not scale well, so some tiny nucleotides in the Logo are unreadable. A higher-resolution image is needed.

Reviewer #2: Comments:

1. Please provide additional information regarding the statistical analysis of the proteomic data provided in Figure 2. Overall, the capsid proteins were detected at low abundance. In contrast, OspC, OspA and GroEL were detected at high abundance. A greater discussion of how these data were analyzed is warranted.

2. Please provide an explanation for why the read coverage of the plasmids shown in Figure 5 is no greater than 29 kb even for plasmids larger than 29 kb.

3. From the conserved motif analysis presented in Figure 8 the conservation of the pac site itself across species is unclear. How much sequence identity is needed for packaging? Is DNA from one species able to be packaged and transduced to another species?

4. Figures 8 and 9 are quite difficult to read. In particular, nucleic acid sequences in Figure 9 are unable to read due to the lack of contrast with the blue and green blocks.

5. Some of the conclusions stated in the Discussion are too strongly worded. Line 347-348—suggests a possible role for, Lines 364-366—also suggests the possibility.

Reviewer #3: 1. The studies were all carried out with strain CA-11.2A, the strain from which φBB-1 was initially isolated. Bacteriophage have been rarely observed with other Bb strains, although they all harbor the cp32 prophage. Further, CA-11.2A produces 20-100 fold more phage than B31 or 297 (the most commonly used Bb laboratory strains). The reason for this remains a mystery. It may be noteworthy that CA-11.2 is an ospC type D strain, a genotype not commonly observed in tick and human isolate studies. Is it possible that this lineage contains some unique sequences that may explain the behavior of CA-11.2A? Perhaps the authors could address this in the Discussion.

2. P. 6 and fig. S1 – The data clearly demonstrate that reads covering the entire lengths of numerous plasmids other than cp32. The authors state that this suggests that full-length versions of these plasmids are packaged. Have the authors detected any reads (contigs) that would be large enough to represent full-length plasmids? It appears that non-cp32 plasmids do not have pac sequences so one would assume that these other DNA elements are packaged in the phage heads via some general transduction mechanism. Could the authors please comment.

3. Fig. S1—Among the full-length plasmids packaged is lp56. This lp56 is quite different from the one found in B31, a portion of which comprises a full-length linearized cp32. The CA-11.2A lp56 is approximately 29 kb in length. The best BLAST matches for CA-11.2A lp56 are to lp56 from strains WI91-23, B331, B500 and B408. These strains all have a similar-sized lp56 and are all ospC type I. CA-11.2A lp56 encodes a PF32 sequence virtually identical to those of these other lp56 plasmids. With regard to the point raised in #1 above, might this strain “similarity” have any implication for the ability to produce greater amounts of phage? Please comment.

4. P.10 – In a beautiful experiment, the authors demonstrate that attaching a cp32 pac site to pBSV2 results in packaging of the shuttle vector. Does this also proceed via a headful mechanism?

5. In Table S1, the results of proteomic analysis of purified virions is presented. In addition to phage capsid proteins, the most abundant proteins are OspA and OspC. Is this simply because these proteins are so abundant in Bb or is there some other explanation?

A few minor editing issues:

6. P.1, Abstract, ln 24 – “.. headful mechanism with preferentially packaging..”. This should read either preferential or delete “with” before “preferentially”.

7. P. 3, ln 94-95 – “re-chloroformed”. Is there such a word in the English language?

8. P. 17, ln 487 --- SM buffer is not defined.

9. p.18, ln509 – “sulfate” should be inserted after ammonium.

10. The Bioproject has illumina reads as well as Nanopore reads. Not sure this matters, but they are not referred to anywhere in the manuscript.

PLOS authors have the option to publish the peer review history of their article (what does this mean?). If published, this will include your full peer review and any attached files.

Reviewer #1: No

Reviewer #2: No

Reviewer #3: No

Figure Files:

Data Requirements:

Reproducibility:

References:

---

## [Editor Report · Decision Letter 1]

13 Mar 2024

Dear Dr. Secor,

We are pleased to inform you that your manuscript 'Characterization and genomic analysis of the Lyme disease spirochete bacteriophage ϕBB-1' has been provisionally accepted for publication in PLOS Pathogens.

Best regards,

John M Leong

Pearls Editor

PLOS Pathogens

David Skurnik

Section Editor

PLOS Pathogens

Michael Malim

Editor-in-Chief

PLOS Pathogens

orcid.org/0000-0002-7699-2064
---

## [Editor Report · Acceptance letter]

26 Mar 2024

Dear Dr. Secor,

We are delighted to inform you that your manuscript, "Characterization and genomic analysis of the Lyme disease spirochete bacteriophage ϕBB-1," has been formally accepted for publication in PLOS Pathogens.

Best regards,

Michael Malim

Editor-in-Chief

PLOS Pathogens

orcid.org/0000-0002-7699-2064